health and disease and epidemiology

Great Barrington Declaration, COVID-19, vaccination, agent-based model, age-stratified interventions, Canada

**Author for correspondence:**
Vanessa Gabriele-Rivet
e-mail: vanessa.gabriele-rivet@canada.ca

†These authors contributed equally to this study.

# Modelling the impact of age-stratified public health measures on SARS-CoV-2 transmission in Canada

Vanessa Gabriele-Rivet[1,†], Kelsey L. Spence[2,†], Nicholas H. Ogden[1], Aamir Fazil[1], Patricia Turgeon[1], Ainsley Otten[1], Lisa A. Waddell[1] and Victoria Ng[1]

[1]Public Health Risk Sciences Division, National Microbiology Laboratory, Infectious Disease Prevention and Control Branch, Public Health Agency of Canada, Guelph, Ontario and St-Hyacinthe, Québec, Canada
[2]Department of Population Medicine, Ontario Veterinary College, University of Guelph, Guelph, Ontario, Canada

VG-R, 0000-0003-0813-7100; NHO, 0000-0002-1062-7283; VN, 0000-0002-7619-541X

Public health measures applied exclusively within vulnerable populations have been suggested as an alternative to community-wide interventions to mitigate SARS-CoV-2 transmission. With the population demography and healthcare capacity of Canada as an example, a stochastic age-stratified agent-based model was used to explore the progression of the COVID-19 epidemic under three intervention scenarios (infection-preventing vaccination, illness-preventing vaccination and shielding) in individuals above three age thresholds (greater than or equal to 45, 55 and 65 years) while lifting shutdowns and physical distancing in the community. Compared with a scenario with sustained community-wide measures, all age-stratified intervention scenarios resulted in a substantial epidemic resurgence, with hospital and ICU bed usage exceeding healthcare capacities even at the lowest age threshold. Individuals under the age threshold were severely impacted by the implementation of all age-stratified interventions, with large numbers of avoidable deaths. Among all explored scenarios, shielding older individuals led to the most detrimental outcomes (hospitalizations, ICU admissions and mortality) for all ages, including the targeted population. This study suggests that, in the absence of community-wide measures, implementing interventions exclusively within vulnerable age groups could result in unmanageable levels of infections, with serious

outcomes within the population. Caution is therefore warranted regarding early relaxation of community-wide restrictions.

# 1. Introduction

Severe acute respiratory syndrome coronavirus 2 (SARS-CoV-2), the pathogen responsible for coronavirus disease (COVID-19), was detected in Wuhan, China in late 2019 [1]. Within less than a year and a half of its detection, SARS-CoV-2 has spread rapidly worldwide, resulting in 146.7 million reported cases and 3.1 million deaths as of 26 April 2021 [2]. With the recent emergence and rapid spread of variants of concern (VOCs), such as the delta variant, associated with increased transmissibility and higher rates of hospitalization, the global threat of COVID-19 has been exacerbated in nearly all countries across the world [3,4]. During the early stages of the epidemic, countries introduced a variety of strategies to reduce transmission, including school and workplace closures, travel restrictions and physical distancing requirements [5]. Over time, these strategies have been re-implemented periodically in response to resurgence of the epidemic, following periods of relaxation [5].

While public health measures (PHMs) have undoubtedly been effective in mitigating SARS-CoV-2 transmission [6,7], restrictive PHMs (closures of businesses, schools, leisure venues, travel restrictions, stay-at-home orders etc.) have very significant negative economic, societal and non-COVID-19 health impacts [8]. An optimal strategy for PHM implementation would be one that balances the need to control COVID-19 with one that minimizes negative health and economic impacts, and societal disruption. Strategies that purport to achieve this have been suggested. These involve lifting community-wide measures in favour of measures targeting vulnerable populations, embodied by the principles of the 'Great Barrington Declaration' [9]. Despite the age distribution of reported cases shifting to younger age groups in several countries [10,11], older individuals continue to be at highest risk for severe outcomes, such as hospitalization and death [12]. Proponents for the age-stratified implementation of PHMs hypothesize that by focusing on protecting only vulnerable populations, while others resume life as normal without any restrictions as a means of gradually achieving herd immunity, the occurrence of severe health outcomes and societal and economic disruption could be minimized [9]. The principles of this declaration have been used to oppose restrictive PHMs and have been considered in policy decisions [13,14].

As in many countries, in Canada, a combination of non-pharmaceutical measures have been used to mitigate SARS-CoV-2 transmission, including the closures of non-essential businesses, cancellation of gatherings and physical distancing policies [15]. Modelling studies have projected that in the absence of sustained community-wide PHMs, the epidemic will resurge [16–18]. However, the impact of relaxing community-wide measures in favour of age-stratified interventions has not been previously examined in Canada. Furthermore, following the approval of several vaccines and their priority allocation to older age groups [19], there is a need to explore the consequences of relaxing community-wide measures while those belonging to vulnerable age groups receive the vaccine.

Epidemiological agent-based models (ABMs) can simulate disease transmission at the individual level, allowing the implementation of unique social and behavioural attributes for each individual (the 'agents') within defined spatial environments. The disease dynamics generated from these models emerge from the accumulation of heterogeneous interactions at the individual level. In general, ABMs are more computationally expensive than other mathematical models, such as compartmental models; however, ABMs allow the simulation of complex dynamic systems [20]. For this reason, model outputs are stochastic and can provide an understanding of a range of different outcomes despite the same model inputs. These models have therefore proven useful to evaluate the impact of public health interventions and policy measures on disease transmission, particularly for response to the COVID-19 pandemic [21,22].

We used an age-stratified agent-based model, based on the demographics of the Canadian population and Canadian healthcare capacity, to test the concept that sheltering vulnerable age groups by shielding them from infection, or by vaccination, allows restrictions to other age groups to be lifted. These theoretical scenarios explored the impact of applying shielding or vaccination exclusively to older age groups under optimal, but unrealistic conditions (i.e. interventions which produce an instant effect and are applied simultaneously to the entire target population with perfect compliance). As such, this experiment was intentionally designed to determine whether the implementation of 'near perfect' age-stratified measures would allow the safe lifting of community-wide PHMs, compared with a scenario with sustained community-wide PHMs applied to all individuals.

# 2. Methods

## 2.1. Model structure

A stochastic age-stratified agent-based model was developed in AnyLogic Professional 8.7.1 to simulate the spread of SARS-CoV-2 in the Canadian population, incorporating both asymptomatic and symptomatic infections. This model has been used to explore PHMs needed to control COVID-19 in Canada and methodological details have previously been published [16,23]. Briefly, the model is a simplified approximation of the general situation occurring across Canada, running on a daily time step over a 2-year period (7 February 2020–7 January 2022). The model includes a population of 100 000 agents, scaled to represent the age structure and demographics of the Canadian population. Agents are each assigned to a household and have an age-specific daily probability of leaving the house to a designated common environment (school, workplace or mixed-age venue) according to their likelihood of spending time in a given location (electronic supplementary material, table S4 in ESM1). Transmission of SARS-CoV-2 occurs within the community through contacts between individuals in any of these environments. Paramatrization of transmission in the model was based on the original wild-type. Further details of the model and its parameter estimates are available in the electronic supplementary material, ESM1.

## 2.2. Benchmark scenario

We used a 'benchmark scenario' to reflect the past and current approaches for using community-wide PHMs to control the epidemic across Canada [23], which are similar to those adopted in other countries [24]. This scenario incorporated the following PHMs, with values informed by Canadian data on mobility, under-reporting and individual compliance (see electronic supplementary material, ESM1 for further details): (i) from 16 March 2020, onwards, a constant rate of 20% of symptomatic cases were detected and isolated with 50% of the detected cases contact traced and quarantined [25–27]. Contact tracing measures were discontinued when the number of active clinical cases exceeded 50 per 100 000 to reflect overcapacity limitations [28,29]; and (ii) from 16 March to 7 September 2020, three phases of community closures and physical distancing were implemented. These phases reflected closures and gradual reopening of schools, workplaces and mixed-age venues over time and varying levels of contact rate reduction outside of the household with age-specific compliance (electronic supplementary material, ESM1) [30,31].

All common environments reopened after 7 September 2020; however, to reflect the past and current intermittent closures implemented across Canadian jurisdictions, shutdowns were activated when the daily number of active clinical cases surpassed 100 per 100 000 people. These shutdowns involved the closure of 100% of schools, 50% of workplaces and 50% of mixed-age venues for a period of 42 days, informed by the initial level of closures that occurred in March 2020. These shutdown events were activated and personal physical distancing was maintained throughout the remaining simulation time. Personal physical distancing was modelled as reducing contact rates to 45% of those pre-COVID-19 with age-specific compliance, based on Canadian survey data on behavioural variations to physical distancing over time [32–34].

## 2.3. Age-stratified intervention scenarios

The age-stratified intervention scenarios tested the implementation of targeted measures for individuals above three age thresholds (greater than or equal to 45, 55 and 65 years old) while lifting various community-wide PHMs. Shutdowns and physical distancing ceased once the age-stratified interventions were applied, whereas minimal control through case detection and contact tracing, as implemented in the benchmark scenario, continued for the duration of the model run. We followed a simplistic assumption that the cessation of community-wide measures and the implementation of age-stratified measures would occur on the same day (1 January 2021). The following interventions were applied, targeting all individuals above the given age threshold: (i) vaccination with an infection-preventing vaccine with a 95% effectiveness, (ii) vaccination with an illness-preventing vaccine with a 95% effectiveness or (iii) shielding from the remainder of the community, without vaccination (table 1). Individuals from the target group were assumed to have perfect compliance to these measures which was maintained across the entire model run.

**Table 1.** Description of the age-stratified interventions explored in the model.

| scenario | description |
| --- | --- |
| infection-preventing vaccine | All individuals above the age threshold are administered an infection-preventing vaccine. Each vaccinated individual has a 95% probability of developing immunity against infection [35,36]. Immune individuals will have a 0% probability of infection given contact with an infected individual and therefore cannot become infected or transmit the infection to others. These individuals maintain regular contacts both inside and outside the household. |
| illness-preventing vaccine | All individuals above the age threshold are administered an illness-preventing vaccine. Each vaccinated individual has a 95% probability of developing immunity against symptomatic infections [35,36]. Immune individuals will have a 0% probability of developing symptoms if infected and are able to transmit the disease to others if infected. These individuals maintain regular contacts both inside and outside the household. |
| shielding | All individuals above the age threshold remain in their household the majority of the time and have a 5% daily probability of leaving the household to visit mixed-age venues. Individuals have reduced contact with all other household members, regardless of their age, by 75%. |

## 2.4. Model outputs

The outcomes of interest included: the overall attack rate, clinical attack rate, asymptomatic attack rate, clinical mortality rate, infection fatality rate, number of hospitalizations, number of intensive care unit (ICU) admissions, number of deaths, number of days over the estimated maximum Canadian hospital bed capacity (30 beds per 100 000) [37] and the estimated maximum ICU bed capacity (9 beds per 100 000) [37]. Due to model stochasticity, each scenario was run for 100 realizations. This number of model realizations was considered appropriate as, in a previous study using this model, 50 realizations were found sufficient to identify the main trends [23]. The median and 95% credible interval (CrI, 2.5th to 97.5th percentiles) of all outputs across each scenario is presented, as well as the percentage change in median values of key metrics between the benchmark scenario and age-stratified intervention scenarios. The smoothed daily median curve and the smoothed 95% credible interval of all scenarios were plotted for three key continuous model outcomes (daily clinical incident cases, daily prevalence in hospitalizations and daily prevalence in ICU). All statistical analyses and creation of plots were performed in R v. 4.0.3 [38].

## 3. Results

Compared with the benchmark scenario, which generated steady epidemic curves across time with a median fluctuating around 10 cases per 100 000 people (electronic supplementary material, figure S1 in ESM2), all age-stratified intervention scenarios, with removal of PHMs, led to a substantial increase in the median attack rate (up to 61.2% attack rate) and an exponential growth in the daily incidence curves (table 2 and figure 1a). Among all age-stratified interventions, the infection-preventing vaccine scenario generated the lowest number of cases, followed by the shielding scenario. For both scenarios, the incidence was further decreased as the age threshold decreased (table 2).

In the benchmark scenario, hospital and ICU bed usage remained under capacity for the model duration, with the exception of a few model realizations (electronic supplementary material, figure S1 in ESM2). By contrast, all age-stratified scenarios resulted in increased hospitalizations, ICU admissions and cumulative deaths (figures 1b,c and 2). Even in the most effective scenarios (vaccine scenarios for people greater than or equal to 45 years old), the median daily hospital and ICU beds exceeded the maximum hospital and ICU bed capacities during the peak of the outbreak for a median of approximately 1.5 to 3 months. Among all age-stratified scenarios, the administration of an infection-preventing vaccine generated the fewest hospitalizations, ICU admissions and deaths for all age thresholds, followed by an illness-preventing vaccine and, finally, the shielding implementation.

**Table 2.** Summary of model output measures presented as the median (95% credible interval) summarizing 100 model realizations for each scenario.

| output | benchmark | infection-preventing vaccine | | | illness-preventing vaccine | | | shielding | | |
|---|---|---|---|---|---|---|---|---|---|---|
| | | age ≥45 | age ≥55 | age ≥65 | age ≥45 | age ≥55 | age ≥65 | age ≥45 | age ≥55 | age ≥65 |
| overall attack rate (%) | 6.5 (5.7–7.5) | 28.3 (2.2–29.5) | 44.7 (2.3–45.6) | 54.7 (2.6–55.7) | 61.2 (2.7–61.9) | 60.6 (2.1–61.6) | 60.5 (2.6–61.2) | 43.3 (2–44.3) | 52.3 (2.9–53.4) | 57.6 (3–58.6) |
| clinical attack rate (%) | 4 (3.5–4.6) | 16.1 (1.3–16.8) | 26.3 (1.4–26.9) | 32.8 (1.7–33.3) | 24.5 (1.4–24.8) | 29.9 (1.3–30.4) | 34 (1.6–34.5) | 25.8 (1.2–26.4) | 31.5 (1.8–32.2) | 35 (1.8–35.7) |
| asymptomatic attack rate (%) | 2.5 (2.2–3) | 12.3 (0.8–12.8) | 18.4 (0.9–18.8) | 21.8 (1–22.3) | 36.7 (1.3–37.4) | 30.6 (0.9–31.3) | 26.4 (1–26.9) | 17.5 (0.8–18.1) | 20.8 (1.1–21.2) | 22.5 (1.1–22.9) |
| clinical mortality rate (%) | 1.4 (1.1–1.8) | 0.4 (0.3–1.8) | 0.6 (0.5–1.5) | 1.2 (0.8–1.4) | 0.5 (0.4–1.3) | 0.7 (0.6–1.5) | 1.2 (0.8–1.7) | 2.1 (1.2–2.3) | 2.1 (1–2.3) | 2.4 (0.9–2.6) |
| infection fatality rate (%) | 0.9 (0.6–1.1) | 0.2 (0.2–1.1) | 0.4 (0.3–0.9) | 0.7 (0.5–0.8) | 0.2 (0.2–0.7) | 0.3 (0.3–0.9) | 0.7 (0.5–1) | 1.3 (0.7–1.4) | 1.3 (0.6–1.4) | 1.4 (0.6–1.6) |
| hospitalizations per 100 000 | 296 (243–357) | 608.5 (100–688) | 1228.5 (98–1318) | 1937 (108–2034) | 958 (93–1038) | 1404 (89–1510) | 2025 (109–2135) | 1633 (92–1757) | 2109 (134–2225) | 2536.5 (137–2682) |
| ICU admissions per 100 000 | 99 (81–125) | 181 (24–209) | 405 (34–451) | 695.5 (39–756) | 278 (30–324) | 466 (30–524) | 739.5 (37–815) | 527 (31–582) | 707 (38–775) | 880 (46–962) |
| deaths per 100 000 | 56 (41–73) | 59.5 (14–78) | 156.5 (15–185) | 377 (17–422) | 115 (17–143) | 182 (15–220) | 412 (18–466) | 540 (16–606) | 667 (21–732) | 823 (22–908) |
| days hospital beds overcapacity | 0 (0–0) | 86 (0–100) | 101 (0–112) | 109 (0–119) | 83 (0–92) | 95.5 (0–105) | 108 (0–118) | 114 (0–126) | 113 (0–124) | 115 (0–123) |
| days ICU beds overcapacity | 0 (0–3) | 51 (0–70) | 88 (0–100) | 99 (0–111) | 67.5 (0–80) | 82.5 (0–98) | 99 (0–110) | 100.5 (0–114) | 102.5 (0–116) | 104 (0–120) |

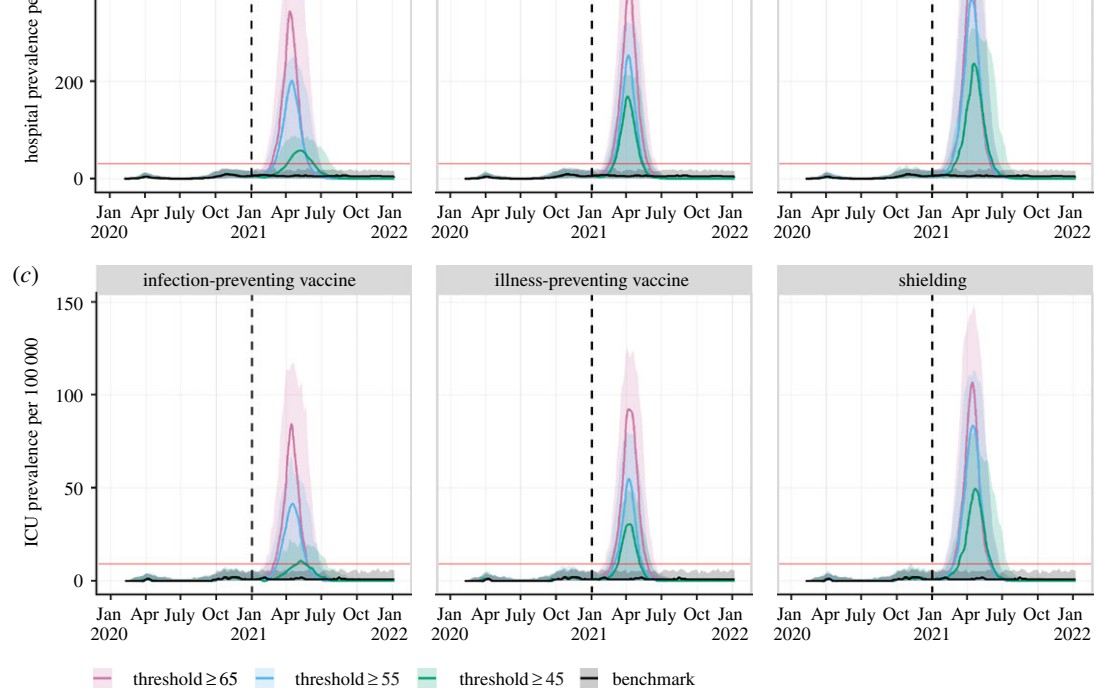

**Figure 1.** Projected epidemic curves showing (*a*) daily incident cases (asymptomatic and symptomatic), (*b*) daily hospital bed prevalence and (*c*) daily ICU bed prevalence per 100 000 people for the benchmark scenario and age-stratified intervention scenarios using different age thresholds (greater than or equal to 45, 55 and 65 years old). For each scenario, the solid curve and shaded area represent the smoothed median and smoothed 95% credible intervals, respectively. The black dashed line represents the date at which the age-stratified intervention was applied. The red solid line in (*b*) and (*c*) represents the estimated Canadian hospital bed capacity (30 per 100 000) and ICU bed capacity (9 per 100 000), respectively.

Examining the outputs by age group revealed that individuals under the age threshold were negatively impacted by the implementation of all age-stratified interventions, as compared with the benchmark scenario, in terms of cases (up to 8-fold increase), hospitalizations (up to 8.5-fold increase), ICU admissions (up to 9-fold increase) and deaths (up to 20-fold increase; table 3 and figure 2; electronic supplementary material, figure S2 in ESM2). In addition to the adverse effect on younger populations, the shielding scenario led to a substantial increase in negative severe outcomes for individuals above the age threshold (up to 5.7-fold and 11.6-fold increase in hospitalizations and deaths, respectively). By contrast, the vaccine scenarios provided some benefit for the vaccinated population; both the illness-preventing vaccine and the infection-preventing vaccine scenarios resulted in reduced hospitalizations and ICU admissions for those above the age threshold. Furthermore, the infection-preventing vaccine

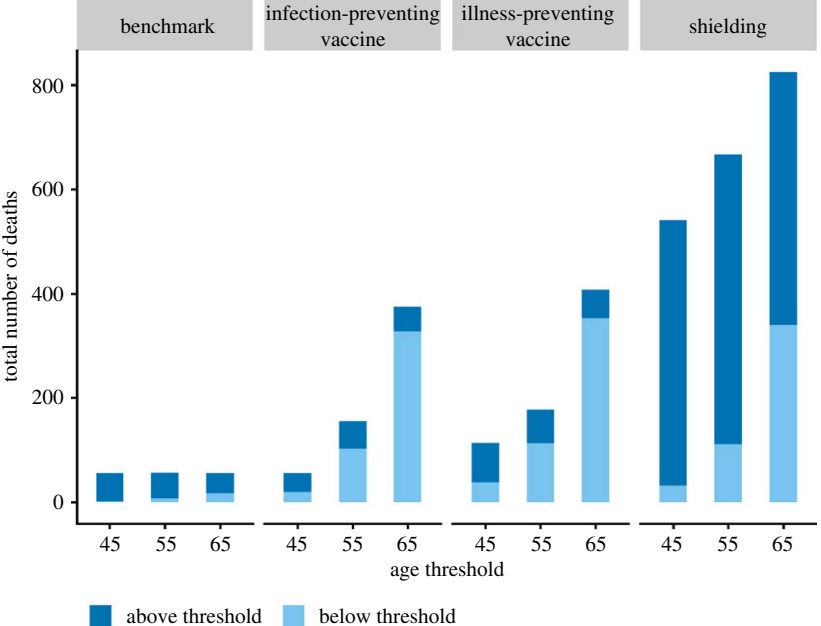

**Figure 2.** Projected cumulative deaths in the benchmark scenario and all age-stratified intervention scenarios (infection-preventing vaccine, illness-preventing vaccine and shielding) stratified by age group (above the targeted age threshold and below the targeted age threshold). In the age-stratified intervention scenarios, the x-axis corresponds to the age threshold for which interventions were applied and the corresponding age stratification for output presentations whereas, in the benchmark scenario, the x-axis is only used to indicate the age stratification for output presentations.

scenario also reduced the number of cases (greater than or equal to 55 and 45 thresholds) and deaths (greater than or equal to 45 threshold) for those above the corresponding age threshold.

## 4. Discussion

This study used an age-stratified agent-based model to explore the impact of lifting PHMs when older individuals are either vaccinated or shielded from the remainder of the population using the Canadian population as an example. Overall, the findings indicate that applying interventions exclusively to these high-risk groups, without sustained PHMs in the community, would generate a resurgence of infections and increased hospitalizations, ICU admissions and deaths. Therefore, if targeted age-stratified interventions are to be considered, PHMs must be maintained in the community to prevent a resurgence of SARS-CoV-2 infections until a sufficiently large proportion of the population is vaccinated. Across all modelled scenarios, the benchmark scenario, which reflects the recurrent community-wide shutdowns as implemented in reality, indisputably offered the safest option. As demonstrated throughout the pandemic, the implementation of restrictive measures in almost all countries across the world, including Canada, has repeatedly proven to be effective in reducing transmission of COVID-19 [39].

Previous arguments for an age-stratified strategy have focused on the benefit of intentionally increasing infections in the younger population in order to build naturally acquired immunity [9]. However, across all age-stratified interventions explored in the present analyses, the younger population experienced considerably worse outcomes compared with the benchmark scenario with sustained community-wide measures, suggesting that this approach would lead to levels of infections among this age group that result in a large number of avoidable deaths. Previous studies have cautioned that aiming to achieve herd immunity through natural infection in the general population, including all age groups, would lead to a substantial increase in deaths [16,40]. According to our analysis, this concept holds true even when attempting to achieve natural herd immunity by only allowing natural infection in the younger and presumably less vulnerable population. In addition to severe outcomes, the health impacts of long-term effects of COVID-19 (Long COVID-19) affect a proportion of COVID-19 cases beyond three months from disease onset; and symptoms range from fatigue and mental health issues to ongoing respiratory complications in individuals that experienced

**Table 3.** Percentage change in median outcome values from the age-stratified intervention scenarios compared with the benchmark scenario (with sustained community-wide public health measures) stratified by age group (above the targeted age threshold and below the targeted age threshold).

| scenario | percentage change in all cases from benchmark scenario (%) | | percentage change in hospitalizations from benchmark scenario (%) | | percentage change in ICU admissions from benchmark scenario (%) | | percentage change in total deaths from benchmark scenario (%) | |
|---|---|---|---|---|---|---|---|---|
| | above threshold | below threshold | above threshold | below threshold | above threshold | below threshold | above threshold | below threshold |
| shielding | | | | | | | | |
| ≥45 | 348.0 | 650.3 | 373.7 | 625.6 | 365.5 | 639.6 | 841.7 | 1500.0 |
| ≥55 | 479.5 | 742.4 | 488.1 | 739.1 | 488.7 | 759.8 | 1023.2 | 1485.7 |
| ≥65 | 603.8 | 793.9 | 571.4 | 831.9 | 590.7 | 860.4 | 1159.7 | 1900.0 |
| illness-preventing vaccine | | | | | | | | |
| ≥45 | 964.4 | 788.5 | −14.1 | 775.0 | −17.6 | 789.6 | 40.7 | 1800.0 |
| ≥55 | 1035.1 | 798.3 | −17.3 | 793.3 | −16.0 | 816.3 | 30.3 | 1514.3 |
| ≥65 | 1123.0 | 811.9 | −10.6 | 858.9 | −14.8 | 893.1 | 41.6 | 1976.5 |
| infection-preventing vaccine | | | | | | | | |
| ≥45 | −45.3 | 484.8 | −47.0 | 464.2 | −48.6 | 468.8 | −32.4 | 850.0 |
| ≥55 | −29.3 | 689.9 | −29.5 | 687.7 | −32.1 | 698.9 | 7.1 | 1357.1 |
| ≥65 | 17.7 | 780.0 | −18.5 | 815.8 | −25.9 | 831.3 | 24.7 | 1823.5 |

mild to severe COVID-19 [41–43]. The long-term health outcomes were not considered in the present analyses, but would represent an additional health burden on younger populations if COVID-19 was allowed to spread uncontrollably in this age group. Considering the overall health risks that would arise from this strategy, the approach of actively promoting the spread of COVID-19 within a portion of the population, even in younger individuals, raises serious ethical concerns.

Our findings suggest that, across all age-stratified scenarios, the approach of shielding older populations while relaxing community-wide PHMs would have the most detrimental impact, leading to a substantial increase in severe outcomes for all age groups including the targeted population, even when using a relatively young age threshold (greater than or equal to 45). Thus, as the epidemic expands in the unrestrained population, the shielding strategy is not sufficient to prevent transmission to the older population, which occurs either within their respective household or during their infrequent visits to common venues (electronic supplementary material, table S1 in ESM2). The lack of positive benefit for the shielding scenario is consistent with other modelling studies that demonstrate that the sole implementation of shielding in the elderly population would not avoid a healthcare system collapse nor reduce deaths [22,44–46]. The ethical implications involved in strictly shielding up to 45% of the population (i.e. with a 45 years old threshold) from the remainder of the population for a prolonged time period also needs to be considered. For instance, along with the direct negative health outcomes resulting from the epidemic, it can be expected that this shielding strategy would eventually lead to severe psychological impacts on the targeted population [47]. Furthermore, the practicalities of implementing such age-stratified interventions in the real world can be questioned, as a significantly large proportion of the workforce would be prevented from working (54% of the workforce in the model are 45 years and older, see electronic supplementary material, tables S1 and S3 in ESM1), which would lead to similar psychological and economic issues as caused by community-wide shutdowns (closure of 50% of workplaces). These ethical and practical negative outcomes would need to be considered in addition to the fact that this age-stratified shielding intervention is considerably ineffective in preventing severe outcomes in the vulnerable population.

Compared with the shielding scenario, the administration of an illness-preventing or an infection-preventing vaccine to older individuals provided a more optimistic picture in terms of reduced severe outcomes in the older population. As expected, these interventions had a protective effect towards the targeted population. As the age threshold was decreased, this protection covered a larger proportion of the population, therefore diminishing the overall healthcare burdens. However, although vaccinated individuals benefited from fewer hospitalizations and ICU admissions, the cumulative deaths in this age group were nonetheless amplified in most vaccine scenarios compared with the benchmark scenario; this results from the overflow of younger individuals into hospitals in the model, leading to an increased mortality rate in hospitalized and ICU patients when the corresponding beds were overcapacity (electronic supplementary material, table S5 in ESM1) reflecting the healthcare system's inability to treat all patients.

Although all explored age-stratified intervention scenarios led to a resurgence once PHMs were lifted, the administration of an infection-preventing vaccine consistently offered the most promising results. Compared with the infection-preventing vaccine scenario, the illness-preventing vaccine yielded slightly worse outcomes across the younger and older age groups. Given that the targeted population in the illness-preventing vaccine remains susceptible to asymptomatic infection and transmission and that asymptomatic infections typically evade detection, an illness-preventing vaccine could in fact contribute to enhancing transmission within the community, including younger populations, as shown in this study. As found in other modelling studies [48], the magnitude of the outbreak following the lifting of PHMs is expected to decrease as the level of protection of the vaccine against transmission increases. The effectiveness of available COVID-19 vaccines after a first and second dose administration is still being evaluated. Cumulative evidence suggests that these vaccines have a combined infection-preventing and illness-preventing effect, with estimates on vaccine effectiveness against infection ranging between 80 and 95% once fully vaccinated (i.e. two doses administered) [49–53]. Given our assumption that the infection-preventing vaccine fully prevents transmission (i.e. a null probability of getting infected for 95% of the population vaccinated), this scenario therefore incorporates the most optimistic infection-preventing characteristics of a highly effective vaccine.

According to our results, the administration of a highly effective infection-preventing vaccine to all individuals aged greater than or equal to 45 years (approx. 45% of the population) is insufficient to safely lift closures and physical distancing restrictions in Canada. Given the current knowledge on the epidemiology of COVID-19, a 45% vaccination coverage is expected to be below the threshold required to achieve herd immunity even with a highly effective vaccine [54]. Consistent with our results, other modelling studies have also found that relaxing PHMs too early during the vaccination

roll-out can lead to detrimental consequences [55,56]. In fact, the anticipated outcomes from the explored scenarios in this study, in which younger Canadian age groups are impacted, are consistent with recent observations. As vaccinations are being administered in the elderly populations, several Canadian jurisdictions have witnessed a concerning rise in infections and hospitalizations among the younger and socially active population, most likely due to a combination of factors including pandemic fatigue and the emergence of new SARS-CoV-2 VOCs with greater transmissibility and virulence than non-VOC lineages [3,4]. Our modelling results in combination with these observations further emphasize the need for caution when relaxing PHMs.

Certain limitations are inherent in the model due to uncertainty on specific data regarding SARS-CoV-2 epidemiology, including transmission probability in asymptomatic individuals and whether the protection following natural infection is long-lasting. The model, which aimed to represent community transmission in the general Canadian population, did not consider transmission in healthcare and long-term facility settings, as these environments would involve specific prevention interventions that differ from the PHMs implemented within the community. Furthermore, although the detection and isolation rates have most likely varied over time and between jurisdictions during the course of the pandemic, data on these specific spatio-temporal changes were unavailable and therefore these rates were kept constant in the model for simplicity. Our model is parametrized using the best available data on the age-specific proportions of cases that require hospitalization care and/or die, which is based on surveillance data and is consistent with observations [57,58] and estimates used in other modelling studies of this type [59].

Additional limitations of this study are associated with the overly optimistic implementation of the age-stratified interventions, including the assumption that interventions are applied on the same date to all individuals, the targeted population has perfect compliance and the immunity acquired from the vaccines develops instantly and persists for the entire model run. While this method of implementing the interventions might be impractical, our findings can be considered as compelling since the incorporation of more realistic strategies (time-varying adherence to interventions, gradual roll-outs, logistical delays, vaccine hesitancy, etc.) would worsen the projected outbreak progression. Furthermore, the emergence of more transmissible and virulent VOCs, as well as VOCs that potentially evade immunity acquired by vaccination or past infection, was not incorporated into the model [3,60]. Given that the inclusion of new VOCs in our simulations would lead to amplified transmission in the population, and increased risk of severe cases, our findings are expected to be conservative. Altogether, even with 'near perfect' interventions applied under the most optimal conditions, the consequences have been found to be devastating. As ideal conditions would never be achieved in the real world, results from this study further substantiates that an age-stratified approach as proposed by the 'Great Barrington Declaration' would not be effective, and could be extremely harmful for the population, at all ages.

# 5. Conclusion

The idea of actively pursuing herd immunity through natural infection while adopting focused protection on vulnerable populations, as proposed by the 'Great Barrington Declaration', seems appealing on the surface; however, our findings suggest that this approach could lead to devastating health impacts, especially in younger age groups, along with a healthcare system collapse. If an age-stratified intervention approach were to be considered, additional community-wide PHMs must be maintained in parallel to avoid detrimental outcomes resulting from SARS-CoV-2 transmission until a sufficiently large proportion of the population has achieved immunity from vaccination. Given the age-stratified priority vaccine allocation, as well as the ongoing emergence of new SARS-CoV-2 variants and their potential impact on the epidemic progression, it is imperative that caution is employed when it comes to early relaxation of PHMs.

Data accessibility. The PHAC COVID-19 code in Anylogic used for the present study is available at the Dryad Digital Repository: https://doi.org/10.5061/dryad.nvx0k6dsm. The data are provided in electronic supplementary material [61].

Authors' contributions. N.H.O. contributed to the conceptualization. A.O., L.A.W. and V.Ng. contributed to the data collection. V.Ng., V.G.-R. and K.L.S. contributed to the model development. K.L.S. and V.G.-R. contributed to the analysis. All authors contributed to interpretation of the data. V.G.-R. and K.L.S. drafted the article. All authors have critically revised the draft and have approved the submitted manuscript.

Competing interests. We declare we have no competing interests.

Funding. We received no funding for this study.

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
