## [Peer Review File · Royal Society Open Science]

Review History

RSOS-210834.R0 (Original submission)

Review form: Reviewer 1

Is the manuscript scientifically sound in its present form?

Yes

Are the interpretations and conclusions justified by the results?

Yes

Is the language acceptable?

Yes

Do you have any ethical concerns with this paper?

No

Have you any concerns about statistical analyses in this paper?

No

Recommendation?

Accept as is

Comments to the Author(s)

Dear authors,

I am pleased to recommend your work for publication. This well-written and well-founded manuscript contribute evidence concerning the contentious subject, the "Great Barrington Declaration". It demonstrates that the approaches proposed in the declaration would not have the desired effects even in ideal scenarios. On the contrary, it would be detrimental to public health, not only to vulnerable populations but all populations.

The weakness of the manuscript is the number of realisations. With only 100 realisations of each scenario, I am not sure that there is enough statistical power to differentiate between outcomes of the scenarios with interventions. Nevertheless, considering the abysmal difference between the baseline and intervention scenarios, I am confident that the conclusion of the manuscript holds strong.

When reading the manuscript, I was left wondering about VOCs until the end of the discussion section. Therefore, I would suggest including a sentence about VOCs at the beginning of the text. It would improve readability and ameliorate the contextualisation of the simulations for any new reader.

Review form: Reviewer 2

Is the manuscript scientifically sound in its present form?

Yes

Are the interpretations and conclusions justified by the results?

Yes

Is the language acceptable?

Yes

Do you have any ethical concerns with this paper?

No

Have you any concerns about statistical analyses in this paper?

No

Recommendation?

Accept with minor revision (please list in comments)

Comments to the Author(s)

In this paper, the authors discuss the possibility of implementing different age-stratified interventions for combating COVID-19 in Canada, by using an agent-based model. I found the paper very timely and interesting. The (numerical) analysis of the system technically sounds and support the findings described in the discussion. However, I have some comments that I believe the authors should clarify:

- In the introduction, I think that the authors could discuss a bit more about the importance of utilizing agent-based models in mathematical and computational epidemiology for testing different interventions. One key drawback with respect to other models (metapopulation or network models) is that analytical results cannot be derived, and the system can only be simulated. On the other hand, the highly detailed granularity of agent-based models can be used to investigate policies that cannot be modeled through metapopulation and network models. I believe that a discussion on this with some references is important for the readers to understand why such types of models are useful.
- In line 103, the authors assume in the model that, from March 2020, 20% of the asymptomatic cases were detected. From the paper, it is not clear if such a fraction remains constant (which seems not very realistic), or if it increases, as testing practices become more widespread and efficacious.
- How is estimated the quantity 45% in lines 116-7?
- How many of the people in the different age cohorts work? I think that this is an important factor to discuss the possibility of implementing age-stratified interventions in the real-world.
- I believe that Section 3, with the discussion of the results, could be expanded, providing some more details on the significance of the findings.
- Similar analyses, in which the efficacy of age-stratified intervention is studied for different countries during the COVID-19 pandemic are present in the literature. Most of them are in agreement with the findings in this paper (see., e.g., Parino et al., Modelling and predicting the effect of social distancing and travel restrictions on COVID-19 spreading, J. R. Soc. Interface, 2021). I believe that the authors should discuss how their findings are related with these works in the literature.
- I also believe that the discussion sections should also contain some more discussion on the real measures that have been implemented in Canada and in other countries, and how the results of these measures can be interpreted in view of the findings presented in this paper. For instance, what is the average retirement age in Canada?

Decision letter (RSOS-210834.R0)

Dear Dr Gabriele-Rivet

On behalf of the Editors, we are pleased to inform you that your Manuscript RSOS-210834 "Modelling the impact of age-stratified public health measures on SARS-CoV-2 transmission in Canada" has been accepted for publication in Royal Society Open Science subject to minor revision in accordance with the referees' reports. Please find the referees' comments along with any feedback from the Editors below my signature.

Please submit your revised manuscript and required files (see below) no later than 7 days from today's (ie 04-Oct-2021) date. Note: the ScholarOne system will 'lock' if submission of the revision

is attempted 7 or more days after the deadline. If you do not think you will be able to meet this deadline please contact the editorial office immediately.

on behalf of Professor Mark Chaplain (Subject Editor)
openscience@royalsociety.org

Reviewer comments to Author:

Reviewer: 1

Comments to the Author(s)

Dear authors,

I am pleased to recommend your work for publication. This well-written and well-founded manuscript contribute evidence concerning the contentious subject, the "Great Barrington Declaration". It demonstrates that the approaches proposed in the declaration would not have the desired effects even in ideal scenarios. On the contrary, it would be detrimental to public health, not only to vulnerable populations but all populations.

The weakness of the manuscript is the number of realisations. With only 100 realisations of each scenario, I am not sure that there is enough statistical power to differentiate between outcomes of the scenarios with interventions. Nevertheless, considering the abysmal difference between the baseline and intervention scenarios, I am confident that the conclusion of the manuscript holds strong.

When reading the manuscript, I was left wondering about VOCs until the end of the discussion section. Therefore, I would suggest including a sentence about VOCs at the beginning of the text. It would improve readability and ameliorate the contextualisation of the simulations for any new reader.

Reviewer: 2

Comments to the Author(s)

In this paper, the authors discuss the possibility of implementing different age-stratified interventions for combating COVID-19 in Canada, by using an agent-based model. I found the paper very timely and interesting. The (numerical) analysis of the system technically sounds and support the findings described in the discussion. However, I have some comments that I believe the authors should clarify:

- In the introduction, I think that the authors could discuss a bit more about the importance of utilizing agent-based models in mathematical and computational epidemiology for testing different interventions. One key drawback with respect to other models (metapopulation or network models) is that analytical results cannot be derived, and the system can only be simulated. On the other hand, the highly detailed granularity of agent-based models can be used to investigate policies that cannot be modeled through metapopulation and network models. I believe that a discussion on this with some references is important for the readers to understand why such types of models are useful.

- In line 103, the authors assume in the model that, from March 2020, 20% of the asymptomatic cases were detected. From the paper, it is not clear if such a fraction remains constant (which seems not very realistic), or if it increases, as testing practices become more widespread and efficacious.

- How is estimated the quantity 45% in lines 116-7?

- How many of the people in the different age cohorts work? I think that this is an important factor to discuss the possibility of implementing age-stratified interventions in the real-world.

- I believe that Section 3, with the discussion of the results, could be expanded, providing some more details on the significance of the findings.

- Similar analyses, in which the efficacy of age-stratified intervention is studied for different countries during the COVID-19 pandemic are present in the literature. Most of them are in agreement with the findings in this paper (see., e.g., Parino et al., Modelling and predicting the effect of social distancing and travel restrictions on COVID-19 spreading, *J. R. Soc. Interface*, 2021). I believe that the authors should discuss how their findings are related with these works in the literature.

- I also believe that the discussion sections should also contain some more discussion on the real measures that have been implemented in Canada and in other countries, and how the results of these measures can be interpreted in view of the findings presented in this paper. For instance, what is the average retirement age in Canada?

===PREPARING YOUR MANUSCRIPT===

If you have been asked to revise the written English in your submission as a condition of publication, you must do so, and you are expected to provide evidence that you have received

language editing support. The journal would prefer that you use a professional language editing service and provide a certificate of editing, but a signed letter from a colleague who is a native speaker of English is acceptable. Note the journal has arranged a number of discounts for authors using professional language editing services (<https://royalsociety.org/journals/authors/benefits/language-editing/>).

===PREPARING YOUR REVISION IN SCHOLARONE===

Author's Response to Decision Letter for (RSOS-210834.R0)

See Appendix A.

Decision letter (RSOS-210834.R1)

Dear Dr Gabriele-Rivet,

I am pleased to inform you that your manuscript entitled "Modelling the impact of age-stratified public health measures on SARS-CoV-2 transmission in Canada" is now accepted for publication in Royal Society Open Science.

COVID-19 rapid publication process:

We are taking steps to expedite the publication of research relevant to the pandemic. If you wish, you can opt to have your paper published as soon as it is ready, rather than waiting for it to be published the scheduled Wednesday.

This means your paper will not be included in the weekly media round-up which the Society sends to journalists ahead of publication. However, it will still appear in the COVID-19 Publishing Collection which journalists will be directed to each week (<https://royalsocietypublishing.org/topic/special-collections/novel-coronavirus-outbreak>).

If you wish to have your paper considered for immediate publication, or to discuss further, please notify openscience_proofs@royalsociety.org and press@royalsociety.org when you respond to this email.

on behalf of Mark Chaplain (Subject Editor)
openscience@royalsociety.org

Appendix A

Dear Professor Mark Chaplain,

The reviewers' comments for the manuscript entitled "*Modelling the impact of age-stratified public health measures on SARS-CoV-2 transmission in Canada*" (Manuscript ID RSOS-210834) have been addressed as described below. For your convenience, changes to the manuscript are identified below, with sentences that have been removed from the manuscript in strikethrough font, and new content is in underlined font. Please note that the line numbers refer to line numbers in the tracked document. We thank you and the reviewers for their constructive comments which have helped to improve the manuscript.

Sincerely,

Vanessa Gabriele-Rivet

Reviewer: 1

Comments to the Author(s)

Dear authors,

I am pleased to recommend your work for publication. This well-written and well-founded manuscript contribute evidence concerning the contentious subject, the "Great Barrington Declaration". It demonstrates that the approaches proposed in the declaration would not have the desired effects even in ideal scenarios. On the contrary, it would be detrimental to public health, not only to vulnerable populations but all populations.

The weakness of the manuscript is the number of realisations. With only 100 realisations of each scenario, I am not sure that there is enough statistical power to differentiate between outcomes of the scenarios with interventions. Nevertheless, considering the abysmal difference between the baseline and intervention scenarios, I am confident that the conclusion of the manuscript holds strong.

When reading the manuscript, I was left wondering about VOCs until the end of the discussion section. Therefore, I would suggest including a sentence about VOCs at the beginning of the text. It would improve readability and ameliorate the contextualisation of the simulations for any new reader.

We would like to thank reviewer #1 for taking the time to review this manuscript and for their positive review.

In a previous published study involving an earlier version of the presented agent-based model, a comparison analyses between a 50-realization model run and a 200-realization model run was conducted (pages 23-25 of Supplementary material from V. Ng, A. Fazil, L. A. Waddell, P. Turgeon, A.

Otten and N. H. Ogden (2021). “Modelling the impact of shutdowns on resurging SARS-CoV-2 transmission in Canada”, Royal Society Open Science). According to this analysis a 200-realization model provided more stable outputs with tighter credible intervals, but a 50-realization model was sufficient to analyse the general trends across the presented scenarios in that study. As pointed out by the reviewer, the scenarios explored in the present study are substantially different to each other and we are therefore confident that 100 realizations are sufficient to support the findings of the manuscript between the different scenarios. We have added text to that effect (lines 160-162):

“This number of model realisations considered appropriate as in a previous study using this model, 50 realisations were found sufficient to identify the main trends [23].

As suggested, we included a sentence in the first paragraph of the introduction on the emergence of VOCs to improve readability and ameliorate the contextualisation of the simulations for readers (lines 42-44):

“With the recent emergence and rapid spread of variants of concern (VOCs), such as the delta variant, associated with increased transmissibility and higher rates of hospitalisation, the global threat of COVID-19 has been exacerbated in nearly all countries across the world [3, 4]”.

And in the first paragraph of the methods (lines 109-110):

“Paramaterisation of transmission in the model was based on the original wild-type.”

Reviewer: 2

Comments to the Author(s)

In this paper, the authors discuss the possibility of implementing different age-stratified interventions for combating COVID-19 in Canada, by using an agent-based model. I found the paper very timely and interesting. The (numerical) analysis of the system technically sounds and support the findings described in the discussion.

We would like to thank reviewer #2 for their time in reviewing our manuscript and for their constructive comments on this study.

However, I have some comments that I believe the authors should clarify:

- In the introduction, I think that the authors could discuss a bit more about the importance of utilizing agent-based models in mathematical and computational epidemiology for testing different interventions. One key drawback with respect to other models (metapopulation or network models) is that analytical results cannot be derived, and the system can only be simulated. On the other hand, the highly detailed granularity of agent-based models can be used to investigate policies that cannot be modeled through metapopulation and network models. I believe that a discussion on this with some references is important for the readers to understand why such types of models are useful.

The following paragraph was added in the introduction to highlight the importance of utilising ABMs in mathematical and computational epidemiology (lines 74-84):

“Epidemiological agent-based models (ABMs) can simulate disease transmission at the individual level, allowing the implementation of unique social and behavioural attributes for each individual (the “agents”) within defined spatial environments. The disease dynamics generated from these models emerge from the accumulation of heterogeneous interactions at the individual level. In general, ABMs are more computationally expensive than other mathematical models, such as compartmental models, however, ABMs allow the simulation of complex dynamic systems [20]. For this reason, model outputs are stochastic and can provide an understanding of a range of different outcomes despite the same model inputs. These models have therefore proven useful to evaluate the impact of public health interventions and policy measures on disease transmission, particularly for response to the COVID-19 pandemic [21, 22].”

- In line 103, the authors assume in the model that, from March 2020, 20% of the asymptomatic cases were detected. From the paper, it is not clear if such a fraction remains constant (which seems not very realistic), or if it increases, as testing practices become more widespread and efficacious.

The 20% represents a constant rate of 20% of symptomatic cases detected and willing to isolate, which was a time-specific estimate based on studies on Canadian under-reporting and compliance to isolation (references 25-27). While this rate has indeed changed over time during the course of the pandemic, the data to estimate time-specific changes are not available so we used a constant rate with adjustments to other parameters in which data was available to capture changes throughout the pandemic. The electronic supplementary material ESM1 contains detailed information on all parameters, including the public health interventions applied in Canada for the baseline model (page 16). To avoid repeating the details already provided in the ESM1, we added the reference to the ESM1 (lines 117-119):

“This scenario incorporated the following PHMs, with values informed by Canadian data on mobility, under-reporting and individual compliance (see electronic supplementary material ESM1 for further details): 1) From March 16, 2020, onwards, a constant rate of 20% of symptomatic cases were detected and isolated with 50% of the detected cases contact traced and quarantined [25-27].”

We also added some discussion on the fact that this rate is constant rather than time-varying in the limitations of the discussion section (lines 310-313):

“Furthermore, although the detection and isolation rates have most likely varied over time and between jurisdictions during the course of the pandemic, data on these specific spatio-temporal changes were unavailable and therefore these rates were kept constant in the model for simplicity.”

- How is estimated the quantity 45% in lines 116-7?

Changes in contact rate reductions (including the 45% estimate from lines 166-167) were estimated from Canadian survey data on physical distancing behaviours through time. Further details on the different physical distancing phases, and the rationale for parameter estimates, are provided in the electronic supplementary material ESM1 on page 16-17. In the text of the manuscript, we added the following information and references to increase clarity in regards to this specific parameter (lines 132-136):

“These ~~activation of these shutdown events~~ were activated and personal physical distancing (45% of regular pre-COVID contact rates with age-specific compliance), was maintained ~~for~~ throughout the remaining simulation time. Personal physical distancing was modelled as reducing contact rates to 45% of those pre-COVID-19 with age-specific compliance, based on Canadian survey data on behavioural variations to physical distancing over time [32-34].”

- How many of the people in the different age cohorts work? I think that this is an important factor to discuss the possibility of implementing age-stratified interventions in the real-world.

This is a very interesting point. To discuss the practicalities of implementing age-stratified interventions in the real-world, a few sentences were added in the discussion (lines 251-258):

“Furthermore, the practicalities of implementing such age-stratified interventions in the real-world can be questioned, as a significantly large proportion of the workforce would be prevented from working (54% of the workforce in the model are 45 years and older, see tables S1 and S3 from the electronic supplementary material ESM1), which would lead to similar psychological and economic issues as caused by community-wide shutdowns (closure of 50% of workplaces). These ethical and practical negative outcomes would need to be considered in addition to the fact that this age-stratified shielding intervention is considerably ineffective in preventing severe outcomes in the vulnerable population.”

- I believe that Section 3, with the discussion of the results, could be expanded, providing some more details on the significance of the findings.

We have expanded the discussion of the results in the Discussion section (section 4) in regards to the following points:

- Limitation related to the constant detection and isolation rate
- The practicalities of implementing age-stratified interventions in the real-world in terms of the proportion of people who will be prevented from working in the shielding scenarios (previous point)
- Comparison between our findings and other studies which have also investigated the impact of age-stratified interventions (next point)
- The positive impact of the shutdowns in the benchmark scenario and how this relates to the efficacy of community-wide restrictions in the real-world (second to next point)

- Similar analyses, in which the efficacy of age-stratified intervention is studied for different countries curing the COVID-19 pandemic are present in the literature. Most of them are in agreement with the findings in this paper (see., e.g., Parino et al., Modelling and predicting the effect of social distancing and travel restrictions on COVID-19 spreading, J. R. Soc. Interface, 2021). I believe that the authors should discuss how their findings are related with these works in the literature.

We have added the reference Parino et al. to our discussion on the comparison between our findings and other studies, which have also investigated the impact of age-stratified shielding interventions (lines 245-246):

“The lack of positive benefit for the shielding scenario is consistent with other modelling studies that demonstrate that the sole implementation of shielding in the elderly population would not avoid a healthcare system collapse nor reduce deaths [22, 42-44].”

The other studies that are referenced in this sentence include:

- Hoertel N, Blachier M, Blanco C, Olfson M, Massetti M, Rico MS, Limosin F, Leleu H. 2020 A stochastic agent-based model of the SARS-CoV-2 epidemic in France. *Nat. Med.* 26, 1417-1421. (doi:<https://doi.org/10.1038/s41591-020-1001-6>)
- Davies NG, Kucharski AJ, Eggo RM, Gimma A, Edmunds WJ, Jombart T, O'Reilly K, Endo A, Hellewell J, Nightingale ES. 2020 Effects of non-pharmaceutical interventions on COVID-19 cases, deaths, and demand for hospital services in the UK: a modelling study. *Lancet Public Health* 5, e375-e385. (doi:[https://doi.org/10.1016/S2468-2667\(20\)30133-X](https://doi.org/10.1016/S2468-2667(20)30133-X))
- Venturieri VR, Gonçalves MS, Fuck VR. 2021 Mitigation of COVID-19 using social distancing of the elderly in Brazil: The vertical quarantine effects in hospitalizations and deaths. *medRxiv*. (doi:<https://doi.org/10.1101/2021.01.12.21249495>)

- I also believe that the discussion sections should also contain some more discussion on the real measures that have been implemented in Canada and in other countries, and how the results of these measures can be interpreted in view of the findings presented in this paper. For instance, what is the average retirement age in Canada?

The following sentences were added at the end of the first paragraph of the discussion section to discuss how real life measures have been implemented in Canada and how the results of these measures can be interpreted in view of the findings in the paper (lines 211-216):

“Across all modelled scenarios, the benchmark scenario, which reflects the recurrent community-wide shutdowns as implemented in in reality, indisputably offered the safest option. As demonstrated throughout the pandemic, the implementation of restrictive measures in almost all countries across the world, including Canada, has repeatedly proven to be effective in reducing transmission of COVID-19 [37].”

In addition, as recommended in the previous point, we have expanded the discussion section to reflect the additional negative impact that the age-stratified shielding intervention would have on the workforce (as well as the psychological issues), compared to the shutdown interventions that have been implemented in the real world. As noted above, we have added the following discussion in the manuscript (lines 251-258):

“Furthermore, the practicalities of implementing such age-stratified interventions in the real-world can be questioned, as a large proportion of the workforce would be prevented from working (54% of the workforce in the model are 45 years and older, see tables S1 and S3 from the electronic supplementary material ESM1) which would lead to similar psychological and economic issues as caused by community-wide shutdowns (closure of 50% of workplaces). These ethical and practical negative outcomes would need to be considered in addition to the fact that this age-stratified shielding intervention is significantly ineffective in preventing severe outcomes in the vulnerable population.”